# Rational Design of SnO_2_ Hollow Microspheres Functionalized with Derivatives of Pt Loaded MOFs for Superior Formaldehyde Detection

**DOI:** 10.3390/nano12111881

**Published:** 2022-05-31

**Authors:** Lanlan Guo, Yuanyuan Wang, Hua Zeng, Yanji Feng, Xueli Yang, Saisai Zhang, Yonghao Xu, Guodong Wang, Yan Wang, Zhanying Zhang

**Affiliations:** 1School of Physics and Electronic Information Engineering, Henan Polytechnic University, Jiaozuo 454003, China; guolanlan@hpu.edu.cn (L.G.); 212011020018@home.hpu.edu.cn (Y.W.); 212111020016@home.hpu.edu.cn (H.Z.); 212111020019@home.hpu.edu.cn (Y.F.); yonghaoxu@hpu.eud.cn (Y.X.); 2Tianjin Key Laboratory of Electronic Materials and Devices, School of Electronics and Information Engineering, Hebei University of Technology, 5340 Xiping Road, Tianjin 300401, China; xlyang@hebut.edu.cn; 3School of Materials Science and Engineering, Henan Polytechnic University, Jiaozuo 454000, China; sszhang@hpu.edu.cn (S.Z.); zhangzy@hpu.edu.cn (Z.Z.); 4The Collaboration Innovation Center of Coal Safety Production of Henan Province, Henan Polytechnic University, Jiaozuo 454000, China

**Keywords:** Pt@Co_3_O_4_, formaldehyde, oxygen deficient, SnO_2_ hollow microspheres

## Abstract

In this work, SnO_2_ hollow microspheres functionalized with different incorporated amounts of Pt@Co_3_O_4_ complex catalyst were innovatively designed by using an MOF template. The results show that sensor based on the optimal incorporated amount of Pt@Co_3_O_4_ not only greatly enhances the response value of SnO_2_ to formaldehyde (R_air_/R_formaldehyde_ = 4240 toward 100 ppm) but also decreases the low detection limit (50 ppb), which is quite outstanding compared with other SnO_2_-based formaldehyde sensors. Further analysis proves that the content of oxygen vacancy and chemisorbed oxygen and the catalytic effect of ultra-small Pt play the key roles in improving the formaldehyde sensing performance. Meanwhile, this present work demonstrates that oxide semiconductors functionalized with the derivatives of MOF templated catalysts may lead to the discovery of new material systems with outstanding sensing performance.

## 1. Introduction

With the improvement of living standards, people are paying more and more attention to air quality. Among the different kinds of volatile organic compounds (VOCs), formaldehyde has been identified as a carcinogen and teratogenic substance by the US Environmental Protection Agency and the World Health Organization [1,2]. When its concentration in air is 0.4–1 ppm, it irritates the eyes, nose and throat. When its concentration continues to increase, exceeding 3 ppm, human cells and tissues are damaged [3,4]. Therefore, the detection of formaldehyde becomes particularly important. At present, infrared absorption spectroscopy, gas chromatography, ultraviolet absorption and chemiluminescence analysis have been developed and used in the detection of formaldehyde [5,6,7,8]. Although these instruments have high accuracy, they are not suitable for formaldehyde detection due to high cost, complex operation and difficult real-time online detection. In recent years, metal oxide semiconductor (MOS) gas sensors have become new hot spots in the research of formaldehyde detection. From a practical view point, MOS sensors need to meet the following key points: (1) high response and low detection limit, which means that the sensor needs to have an obvious response to low concentrations of formaldehyde; (2) high selectivity, which indicates that the sensor needs to have a specific recognition function for formaldehyde; (3) good stability, representing stable output capacity, which determines the effective life of the sensor. 

Tin oxide (SnO_2_), as a typical n-type semiconductor, is demonstrated to have excellent sensing properties for formaldehyde. However, pure SnO_2_ sensing material still has many shortcomings, such as low response value, unsatisfactory detection limit and poor selectivity. In traditional approaches, noble metals, such as Pd, Au, Ag and Pt, incorporated into SnO_2_ have been synthesized and the gas sensing properties toward formaldehyde have been evaluated. For instance, Ruan’s group prepared Pd-decorated hollow SnO_2_ nanofibers [9]. The results of sensing performance showed that the SnO_2_ modified by Pd had high response to formaldehyde (18.8 toward 100 ppm) and low detection limit (5 ppm) at 160 °C. In Chung et al.’s work, a sensor based on a Au@SnO_2_ core–shell structure was successfully built to detect formaldehyde gas [10]. The response value to 50 ppm formaldehyde was about 11 at room temperature. Dong and his colleagues also found that sensors based on Ag-decorated SnO_2_ possessed enhanced response (14.4 toward 10 ppm) to formaldehyde compared with the pure SnO_2_ [11]. Furthermore, sensors based on Pt incorporated into SnO_2_ were also fabricated to detect formaldehyde in previous work [12,13]. They showed a higher response and lower detection limit to formaldehyde which was superior to Pd, Au and Ag-SnO_2_. Furthermore, apart from SnO_2_ sensing materials, Pt had also been incorporated in other metal oxides when formaldehyde was selected as the target gas. For example, Dong et al. also found that sensors based on Pt-functionalized NiO possessed enhanced response to formaldehyde compared with the pure NiO [14]. Ultra-fast and highly selective room-temperature formaldehyde sensors have also been obtained by Pt-loaded MoO_3_ nanobelts in Gu et al.’s work [15]. Hence, it seems that the incorporation of Pt can indeed improve the sensing characteristics of SnO_2_ sensors for detecting formaldehyde. However, it is worth noting that the sizes of Pt nanoparticles obtained in the literature above are mostly greater than 5 nm. In general, the catalytic activity increases drastically as the particle size of noble metals decreases, leading to the significantly enhanced response. Therefore, finding an effective method to obtain ultra-small Pt nanoparticles and then to enhance the formaldehyde sensing properties of SnO_2_ is imperative.

Cobalt oxide (Co_3_O_4_), as a typical p-type semiconductor, is an attractive functional material in the field of gas sensing. Importantly, building a p-n heterojunction between Co_3_O_4_ and an n-type MOS has attracted considerable interest in improving the sensing characteristic. Liu et al. prepared Co_3_O_4_/ZnO composites via the wet-chemistry route and the response reached 20 toward 100 ppm formaldehyde at an operating temperature of 180 °C [16]. Zhang et al. fabricated Co_3_O_4_/In_2_O_3_ ribbon for detecting formaldehyde [17]. The as-prepared sensing materials displayed a high sensing response of 39 toward 200 ppm formaldehyde at 260 °C. However, incorporation of Co_3_O_4_ into SnO_2_ to form a hetero-structure for formaldehyde sensing applications is rarely reported.

Inspired by the above, in order to further improve the sensing characteristics of formaldehyde, a novel material system of SnO_2_ containing Co_3_O_4_ and ultra-small Pt nanoparticles should be developed. In this work, an effective catalyst loading method by using MOF templated catalyst (Pt@ZIF-67) was adopted. TEM results indicated that fine Pt nanoparticles of ~3 nm were uniformly encapsulated in ZIF-67 without agglomeration. Then, Pt@Co_3_O_4_ complex catalysts were unprecedentedly functionalized on SnO_2_ hollow microspheres through a solvothermal method followed by calcination. Consequently, as expected, an ultra-sensitive formaldehyde sensor based on Pt@Co_3_O_4_-SnO_2_ hollow microspheres was achieved. 

## 2. Materials and Methods

### 2.1. Preparation of ZIF-67

The preparation process of ZIF-67 was as follows. Firstly, Co(NO_3_)_2_·6H_2_O (582 mg) and 2-methylimidazole (656.8 mg) were added into 25 mL methanol solvent under stirring, respectively. After ten minutes, the two kinds of solution were mixed together. Next, the mixed solution continued to be stirred for half an hour and then aged for one day. The precipitate was collected by centrifugation (6000 r/min, 5 min) and washed with ethanol and dried at 60 °C for 15 h.

### 2.2. Preparation of Pt@ZIF-67

Pt@ZIF-67 was prepared by adopting an improved method as stated in reference [18]. ZIF-67 (40 mg) and (NH_4_)_2_PtCl_6_ (4 mg) were dispersed in 3 mL deionized water and stirred for six hours. During this process, platinum ions would enter the cavity of ZIF-67. Subsequently, sodium borohydride (1 mg/mL) solution was applied to reduce Pt^4+^ to Pt^0^. Accordingly, ZIF-67 templated Pt (Pt@ZIF-67) nanoparticles were successfully collected by centrifugation (8000 r/min) and washed six times with deionized water and ethanol alternately, then dried in air at 60 °C for 20 h.

### 2.3. Preparation of Pt@Co_3_O_4_-SnO_2_ Hollow Microspheres

The composites of Pt@Co_3_O_4_-SnO_2_ hollow microspheres were prepared via a hydrothermal method. SnCl_4_·5H_2_O (701.2 mg), oxalic acid (63 mg) and an appropriate amount Pt@ZIF-67 (3 mg, 6 mg and 9 mg, respectively) were added into 9 mL deionized water and 16 mL glycerol. After stirring for 15 min, the mixed solution was transferred into a 50 mL stainless steel autoclave lined with Teflon and kept at 160 °C for 12 h in an oven. After the temperature of the oven dropped, the precipitate could be collected by centrifugating (8000 r/min) and washing six times with deionized water and ethanol alternately, and then dried at 60 °C. Ultimately, the precursor was annealed in air at 500 °C for 2 h with a heating rate of 2 °C/min to obtain Pt@Co_3_O_4_-SnO_2_ hollow microspheres. The obtained composites were denoted as S3 (3 mg Pt@ZIF-67), S4 (6 mg Pt@ZIF-67) and S5 (9 mg Pt@ZIF-67).

Co_3_O_4_@SnO_2_ hollow microspheres (labeled as S2) were prepared through the same experimental procedure of S4 samples, except that the same amount of ZIF-67 was used instead of Pt@ZIF-67 for preparing precursor solution. 

The preparation steps of pure SnO_2_ hollow microspheres (labeled as S1) were the same as S2–S5 except that Pt@ZIF-67 or ZIF-67 was not added.

### 2.4. Material Characterization

The morphologies and surface characteristics of these samples were observed by scanning electron microscopy (SEM, Merlin Compact, Carl Zeiss NTS GmbH, Oberkochen, Germany). The crystallinity, purity and composition were examined by X-ray diffraction (XRD, Smartlab, Rigaku Corporation, Tokyo, Japan) with Cu-Kα1 radiation (λ = 0.15406 nm). The internal microstructures, crystal planes and element distribution of the samples were analyzed by transmission electron microscope (TEM, Titan G260-300, FEI, Hillsboro, OR, USA), high-resolution TEM (HRTEM) and energy-dispersive X-ray spectroscopy (EDS). Furthermore, the research of oxygen species in the samples and the study of valence states of Pt, Co and Sn were carried out by X-ray photoelectron spectroscopy (XPS, ESCALAB 250Xi, Thermo Scientific K-Alpha, Waltham, MA, USA) with a mono Al K_α_ (1486.6 eV, 6 mA × 12 kV).

### 2.5. Measurement of Formaldehyde Gas Sensors

In this work, ceramic substrate with attached silver–palladium interdigitated electrodes (Ag-Pd IDEs) is adopted, whose specific parameters are shown in Figure 1a. The specific steps of the fabrication of the sensor devices are as follows to ensure that the consistency of the resistance baseline and response of multiple sensors are the same. An appropriate amount (3–5 mg) of the as-prepared samples is mixed with a certain amount (0.3–0.5 mL) of ethanol to form a uniform slurry. The slurry is then applied to the surface of the Ag-Pd IDEs with a small brush to form a dense sensing layer. Figure 1b shows the schematic illustration of the blank sensor coated with the sensing layer. Furthermore, the multifunctional gas sensing testing system, CGS-MT (Beijing Sino Aggtech, Beijing, China), which is composed of a temperature control system, test chamber (5 L) with two sensor channels (Figure 1c) and circulating water system, is used to analyze the gas sensing performance of the as-obtained samples. The data acquisition is achieved through test software on the computer connected to the CGS-MT instrument. The sensing performance is studied under laboratory conditions (~15 RH%, 18 °C).

The corresponding gas concentration (e.g., formaldehyde) can be derived from the following Equation [19]:(1)$$C=\frac{22.4\times \phi \times \rho \times V_{1}}{M\times V_{2}}\times 1000$$
where *C* (ppm) and *M* (g/mol) are the concentration and the molecular weight of formaldehyde, respectively. *φ* and *ρ* (g/mL) represent the mass fraction (37%) and the density (0.82 g/mL) of the formaldehyde aqueous solution, respectively. *V*_1_ (μL) and *V*_2_ (5 L) are the volume of formaldehyde aqueous solution to be injected and the volume of the test chamber, respectively.

The details of the complete test procedure are as follows. First, open the circulating water system and put the fabricated sensors into an air-filled test chamber for aging for three days. Afterwards, the sensor resistances gradually tend to a stable value, usually represented by *R_a_*. Second, inject a certain concentration of target gas into the closed chamber with a microliter syringe. After a while, the resistance stabilizes again and is abbreviated as *R_g_*. Then, open the lid of the chamber to refill the entire chamber with fresh air, and the resistance returns to near *R_a_* again. During the test process, the resistance and response change continuously with time and are recorded by the test software on the computer connected to the CGS-MT instrument. The response value and response/recovery times are defined as follows. Taking the n-type MOS sensor to detect reducing gas as an example, the response value is defined as *R_a_*/*R_g_*. The response (*τ_res_*) and recovery (*τ_recov_*) times refer to the time periods from *R_a_* changing to *R_a_* − |*R_a_* − *R_g_*| × 90% and from *R_g_* changing to *R_g_* + |*R_a_* − *R_g_*| × 90%, respectively [15]. 

## 3. Results

### 3.1. Structural and Morphological Characteristics

Figure 2a shows the XRD diffraction patterns of ZIF-67, Pt@ZIF-67 and simulated ZIF-67. The diffraction peaks of as-synthesized ZIF-67 and Pt@ZIF-67 are identical to the results of the published XRD data of simulated ZIF-67 [20]. On the one hand, it indicates that the ZIF-67 synthesized in this work is pure phase. On the other hand, it shows that the encapsulation of noble metal Pt does not change the internal structure of ZIF-67. Figure 2b,c present the SEM images of ZIF-67 and Pt@ZIF-67, respectively. As can be seen, the morphologies of Pt@ZIF-67 do not match well with ZIF-67. Compared with the solid structure of ZIF-67 (Figure 2d), Pt@ZIF-67 shows an obvious core–shell structure (Figure 2e), indicating that the encapsulation of Pt into ZIF-67 destroys the original morphology of ZIF-67 [21,22]. The reason can be explained as follows. During the process of the Pt^4+^ entering the ZIF-67 cavity, the NH^4+^ coming from (NH_4_)_2_PtCl_6_ hydrolyzes to produce H^+^, which provides an acidic environment that can corrode and dissolve ZIF-67 from the outside to inside (NH_4_^+^ + H_2_O → NH_3_·H_2_O + H^+^). Similar results appear in our previous work [20]. Meanwhile, the conclusion that ZIF-67 can be corroded and dissolved in an acidic environment has also been confirmed in the literature [22,23]. From Figure 2e, it can also be concluded that large numbers of dark nanoparticles of about 3 nm are evenly distributed on ZIF-67. Figure 2f proves that the interplanar spacing of well-distributed dark nanoparticles is 0.225 nm, which exactly corresponds to that of Pt (111) [24].

Figure 3a reveals the XRD diffraction patterns of the five samples. All the recorded diffraction peaks of S1–S5 are well indexed to the characteristic peaks of SnO_2_ (JCPDS: 77–447). It is remarkably observed that there are no characteristic peaks of Co_3_O_4_ or Pt in the XRD patterns, which is mainly due to their relatively small incorporated amounts. Importantly, the influence of the derivatives of ZIF-67 (Pt@ZIF-67) on the phase structures of SnO_2_ hollow spheres has been found from the 25.5° to 27.5° amplified diffraction peaks (Figure 3b). It seems that the derivatives of ZIF-67 (Pt@ZIF-67) do not cause any shift in diffraction peaks due to the ultra-low incorporated concentration. 

The SEM results of S1–S5 are shown in Figure 4a–e. It can be clearly seen that the as-prepared samples are highly uniform with a hollow sphere-like morphology, and the diameters are about 2 μm. Moreover, from the high-magnification SEM of the inserted images of S1–S5, the nanoparticles on the surface of S2–S4 samples are more dense and smaller compared to S1. In addition, the low-magnification panoramic SEM image of S4 is given in Figure 4f, which not only indicates the good dispersity of these microstructures, but also confirms the porous and hollow property again.

The TEM image of Figure 5a clearly shows the hollow structure of S1 (pure SnO_2_), which is consistent with the experimental result of the SEM. The inset in Figure 5a is the selected area electron diffractive (SAED) pattern, confirming that pure SnO_2_ is polycrystalline in nature. The HRTEM result in Figure 5b reveals the lattice distances of 0.177 nm, 0.264 nm and 0.335 nm which correspond to the crystal plane of SnO_2_ (211), (101) and (311), respectively. Figure 5c,d show the relevant TEM and HRTEM images of S4. Obviously, the porosity of the shell of S4 is significantly lower than that of S1, which is also consistent with the result of the SEM images. The EDS elemental mapping images exhibited in Figure 5e–h confirm the presence of Sn, Co, O and Pt elements in the S4 samples. It is worth noting that Pt and Co exist in the section shown by the white rectangle, indicating that part of ZIF-67 has not been corroded and dissolved during the stirring and solvothermal process, while the Pt in the yellow rectangle is dispersed in SnO_2_ hollow microspheres in the form of ultra-small nanoparticles. Additionally, it can be clearly seen that other cobalt element are uniformly distributed in SnO_2_ hollow microspheres. In summary, both platinum and cobalt exist in two forms, one of them in which both are loaded on SnO_2_ in the form of Pt@Co_3_O_4_ complex catalyst, and the remaining platinum and cobalt are incorporated into the SnO_2_ hollow microspheres in the form of nanoparticles and doping.

The chemical states of Sn, Co and Pt in S1–S5 are verified by the high-resolution XPS spectra in Figure 6a–c. Figure 6a shows the two characteristic peaks of Sn 3d at ~494.3 eV and ~486.1 eV for Sn 3d_3/2_ and Sn 3d_5/2_, respectively, suggesting that the chemical state of the Sn element is +4 [25,26]. Figure 6b shows that the valence states of Co in the samples are composed of +2 and +3. In particular, the binding energies of Co^2+^ peaks are at ~780.3 eV and ~794.8 eV, and the binding energies corresponding to Co^3+^ are ~776.0 eV and ~793.8 eV [27]. Figure 6c displays the valence states of Pt [28,29]. For the XPS peaks of Pt in S3 and S4, they are not obvious due to the ultra-low incorporated amount. Therefore, the high-resolution XPS peak of platinum in S5 is studied. As shown in Figure 6c, peaks of Pt 4f_5/2_ and Pt 4f_7/2_ are located at 73.3 and 70.0 eV (gap = 3.3 eV), respectively, illustrating the nature of metal platinum (Pt^0^). There is no other obvious peak when the binding energy is greater than 75 eV, which also proves that Pt^2+^ and Pt^4+^ mostly do not exist in the sample [30,31,32]. Figure 6d reveals the high-resolution O 1s XPS analysis of S1–S5. The O 1s spectra can be classified into three oxygen species, named O1, O2 and O3, with binding energy of ~529.5, ~530.5 and ~532.0 eV, respectively. O1 is attributed to the lattice oxygen. O2 represents the hydroxyl groups bonded to the metal cations in the oxygen vacancy region. O3 corresponds to chemisorbed and dissociated oxygen from the water molecules unavoidably adsorbed on the surface [33,34,35,36]. Among them, O1 is relatively stable and does not have any influence on the response. O2, as active sites, can support the adsorption and chemical reaction of oxygen and formaldehyde gas, proving that an increase in O2 content can indeed enhance the response of the sensor [37,38,39]. O3, namely O^−^(ads), can directly participate in the oxidation–reduction with formaldehyde. Obviously, the total relative percentages of O2 and O3 in S4 have been greatly improved due to the incorporation of an appropriate amount of Pt-Co_3_O_4_ in SnO_2_ hollow microspheres, whose value has reached 27.5%.

### 3.2. Gas Sensing Characteristics

Firstly, the response influenced by the operating temperature is investigated. As shown in Figure 7a, S4 exhibits the best response to formaldehyde (R_air_/R_formaldehyde_ = 4240 toward 100 ppm), which is 10.4 times that of pure SnO_2_ (408 toward 100 ppm). Within the temperature range of 200 to 350 °C, the optimum operating temperature of all sensors based on S1–S5 for detecting formaldehyde is the same, that is, 300 °C. Moreover, the incorporated amount of Pt@Co_3_O_4_ has a significant effect on the response of SnO_2_ toward formaldehyde. Specifically, when the amount of Pt@Co_3_O_4_ incorporated is increased in S4 from that in S3, the response of SnO_2_ toward formaldehyde is enhanced from 2990 to 4240. When the incorporated amount of Pt@Co_3_O_4_ continues to increase in S5, the response decreases instead, which is may be due to the saturation of catalytic sensitization and catalyst aggregation caused by the excessive amounts of noble metal Pt [40]. Figure 7b shows the response of the five sensors based on S1–S5 with respect to different gases at a concentration of 100 ppm at 300 °C, proving that all sensors, especially the S4 sensor, have good specific detection capabilities for formaldehyde. Figure 7c demonstrates the superior reliability of the five sensors by evaluating the transient response characteristics toward six cycles of 100 ppm formaldehyde at 300 °C. In addition, to optimize the sensing performance of the sensors at 300 °C, the explicit relationship between formaldehyde concentration and response is further investigated in detail. As shown in Figure 7d, the S4 sensor always shows the highest response toward 1–150 ppm of formaldehyde among the five sensors. The inset figure is the responses of the sensors toward low concentration (1–10 ppm) of formaldehyde.

Subsequently, other sensing characteristics of the S4 sensor toward formaldehyde are further shown in Figure 8. First of all, compared with the pure SnO_2_ hollow microspheres (S1), the sensor based on S4 not only shows a higher response, but also shorter response time. As shown in Figure 8a, the response times of S1 and S4 toward 100 ppm formaldehyde at 300 °C are 26 s and 13 s, respectively, implying that SnO_2_ hollow microspheres functionalized with an optimal amount of Pt@Co_3_O_4_ are a key factor in achieving both an excellent response and response time. Secondly, the dynamic resistance variation curve of the S4 sensor toward 1–0.05 ppm of formaldehyde at 300 °C is exhibited in Figure 8b. The response of the S4 sensor toward 1 ppm formaldehyde at 300 °C is 16.5. Notably, the result reveals a noticeable response to a low concentration of formaldehyde and the detection limit is 50 ppb (R_air_/R_formaldehyde_ = 1.9). Moreover, long-term stability of response and resistance of the S4 sensor for detecting 100 ppm formaldehyde at 300 °C is observed. As shown in Figure 8c, the results show that the attenuation rates of the response and resistance are 11% and 24%, respectively, indicating that the sensor based on S4 is of great significance to realize the accurate detection of formaldehyde gas. Importantly, the comparisons of response and limit of detection between the S4 sensor and other reported SnO_2_-based formaldehyde sensors are displayed in Table 1 [9,12,13,41,42,43,44,45,46,47]. Mostly, the sensor fabricated in this work has outstanding advantages in terms of response and limit of detection. It is worth noting that the limits of detection of [12] (60 ppb) and this work (50 ppb) are not greatly different. Compared with the doping amount (1 wt%) and particle size of Pt (>5 nm) in [12], the sensing properties of Pt@Co_3_O_4_-SnO_2_ toward formaldehyde are significantly enhanced even by the quite low Pt loading (~0.3 wt%) due to the finer dispersion of Pt (~3 nm) attained by using Pt@ZIF-67. Therefore, the sensor fabricated in this work is more meaningful for eventual practical application due to the expensive price of noble metals.

## 4. Discussion

SnO_2_ is one of the most widely researched n-type MOS materials, whose sensing mechanism is related to the adsorption/desorption of formaldehyde on the surface of sensitive materials and controlled by the chemical reaction between formaldehyde and chemically adsorbed oxygen [48,49]. Generally, for sensors based on SnO_2_ in air, the oxygen molecules adsorbed (O_2_(ads)) on the surface of SnO_2_ will remove electrons from the conduction band of SnO_2_, thereby resulting in a decrease in the numbers of carriers in SnO_2_. Then, the sensor remains in a high-resistance state (*R_a_*). Meanwhile, the oxygen will become different kinds of chemically adsorbed oxygen (mainly O^−^ (ads), O^2−^ (ads) and O^−^_2_ (ads)), whose generating process can be expressed by the formulas below [50].
O_2_ (gas)→O_2_ (ads),(2)
O_2_ (ads) + e^−^→O^−^_2_(ads) (T < 150 °C),(3)
O_−_2__ (ads) + e^−^ →2O^−^ (ads) (150 °C < T < 400 °C),(4)
O^−^ (ads) + e^−^ →O^2−^(ads) (T > 400 °C).(5)

Considering the formulas above and the optimal temperature (300 °C) in this work, the chemically adsorbed oxygen mainly reacts with formaldehyde in the form of O^−^ (ads). The reaction equation is as follows [51].
HCHO(ads) + 2O^−^ (ads) → CO_2_ + H_2_O + 2e^−^.(6)

Accordingly, the generated electrons in the chemical reaction will return to the conduction band of SnO_2_, increasing the number of the carriers, thus leading to the sensor having a low-resistance state (*R_g_*). According to the formula of response and response times, the specific value can be calculated. In this work, the enhanced formaldehyde sensing performance of S4 can be attributed to three main reasons. 

(I) The large difference in charge carrier concentration in Pt@Co_3_O_4_-SnO_2_ and SnO_2_ plays a leading role in improving the formaldehyde sensing performance, which can be explained by the change in the initial resistance of the sensor in air (*R_a_*). Figure 9 provides the resistances in air of the five sensors. The *R_a_* of the sensor based on S1 is 8.6 MΩ which is enhanced 28 times with Pt@Co_3_O_4_ incorporation (S4, 242 MΩ). This is closely related to the existence of p-type Co_3_O_4_ and ultra-small Pt nanoparticles. On one hand, p-n heterojunctions between p-Co_3_O_4_/n-SnO_2_ can contribute to the enhancement of the resistance. The work function of p-Co_3_O_4_ (6.1 eV) is larger than that of n-SnO_2_ (4.91 eV) [52,53]. When the p-Co_3_O_4_ grain encounters the n-SnO_2_ grain, a p-n heterojunction is produced at the interface, causing the electrons in SnO_2_ to be transferred to Co_3_O_4_ and the holes in Co_3_O_4_ to move to SnO_2_. Thus, a potential barrier will be created with energy band bending, directly increasing the R_a_ value of the sensor. On the other hand, the spillover and catalytic effect of ultra-small Pt play important roles in regulating the *R_a_* value. It is well known that Pt can act as a specific site which is conducive to making the oxygen molecules in air easily adsorb and then transfer to chemisorbed oxygen, thus significantly increasing the amount of O^−^(ads), which can be confirmed by the high-resolution XPS spectra of O 1s (Figure 10) [54]. As shown in Figure 6d, the relative content of O3 in S4 is the highest among the five samples, being consistent with the theoretical analysis above. According to the formula in (4), this process will cause more electrons to be extracted from the conduction band of SnO_2_. Therefore, the *R_a_* of SnO_2_ further increases with the introduction of Pt. Since the sensor response is defined as the ratio of *R_a_* to *R_g_* (*R_a_*/*R_g_*), a better response is usually accompanied by a higher initial resistance. Thus, it is reasonable that a sensor based on SnO_2_ functionalized with Pt@Co_3_O_4_ has a more excellent response. However, the response change trend of the S5 sensor is inconsistent with the change in *R_a_*, suggesting that it is not comprehensive enough to consider the influence of *R_a_* on response alone. As is well known, the incorporation of excessive noble metals will make the surface catalytic activity of the sensitive body excessively enhanced, resulting in the desorption rate of formaldehyde molecules being greater than the adsorption rate [55]. Consequently, fewer formaldehyde molecules will participate in the reaction with chemically adsorbed oxygen, and finally the response will be reduced instead.

(II) The increase in oxygen vacancy is considered to be another key reason to improve the response of the sensor. As shown in Figure 6d, it can be concluded that the percentage of oxygen vacancy (O2) in the S4 sample is also the highest. By combining that with the analysis above, it makes sense that the incorporation of an optimal amount of Pt@Co_3_O_4_ can maximize the response of SnO_2_ toward formaldehyde. 

(III) The improved response times of the S4 sensor can be understood in relation to the catalytic effect of ultra-small Pt nanoparticles [56]. Due to the ultra-small Pt, formaldehyde molecules can be easily decomposed into many active groups, which will promote the sensing reaction between formaldehyde and chemically adsorbed oxygen (O^−^(ads). As a consequence, the response time value effectively decreases when Pt is incorporated into the SnO_2_ hollow microspheres.

## 5. Conclusions

In this work, we have prepared series of Pt@Co_3_O_4_-SnO_2_ hollow microspheres by using different incorporated amounts of Pt@ZIF-67. Compared with pure SnO_2_ (S1), the sensor based on Pt@Co_3_O_4_-SnO_2_ hollow microspheres (S4) showed the most excellent response to formaldehyde (S = 4240 toward 100 ppm), which is 10.4 times that of pure SnO_2_ (S1). Furthermore, the response time and selectivity are also enhanced after the incorporation of Pt@Co_3_O_4_ into SnO_2_. These noticeably improved sensing properties are achieved by the p-n heterojunction between p-Co_3_O_4_ and n-SnO_2_, the spillover and catalytic effect of ultra-small Pt and the increase in oxygen deficient. These results prove that the unique synthetic route of Pt@Co_3_O_4_ complex catalyst incorporated into MOS is a highly effective way to fabricate chemiresistive sensors.

## Figures and Tables

**Figure 1 nanomaterials-12-01881-f001:**
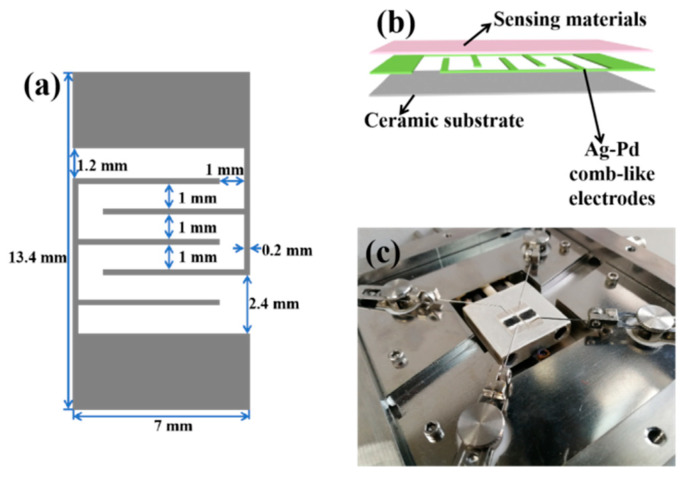
Schematic structure of the blank sensor (**a**) and the fabricated sensor (**b**). (**c**) The test chamber of the gas sensing equipment.

**Figure 2 nanomaterials-12-01881-f002:**
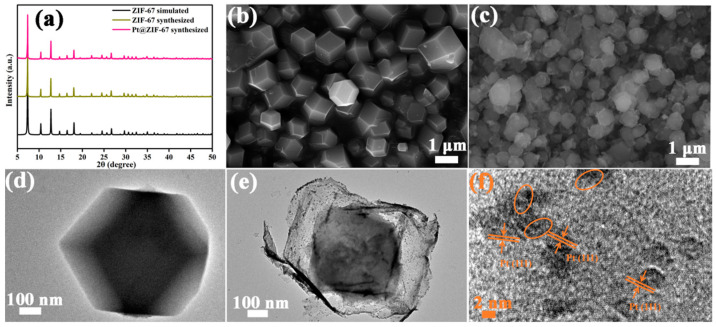
(**a**) XRD patterns of ZIF-67 and Pt@ZIF-67. (**b**,**c**) SEM images of ZIF-67 and Pt@ZIF-67. (**d**) TEM image of ZIF-67. (**e**,**f**) TEM and HRTEM images of Pt@ZIF-67.

**Figure 3 nanomaterials-12-01881-f003:**
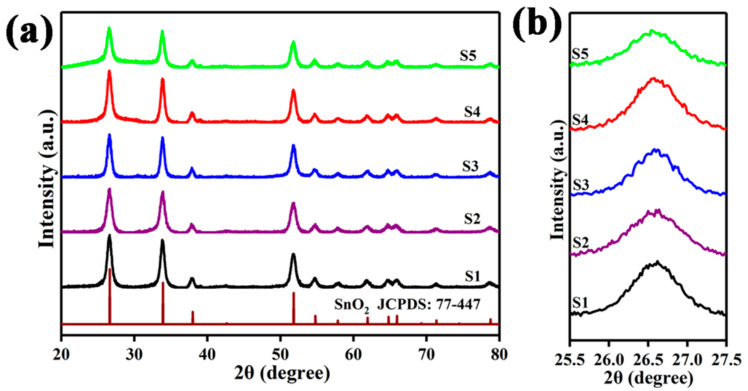
(**a**) XRD analysis of S1–S5. (**b**) The close spectrum of diffraction peaks between 25.5° and 27.5° in S1–S5.

**Figure 4 nanomaterials-12-01881-f004:**
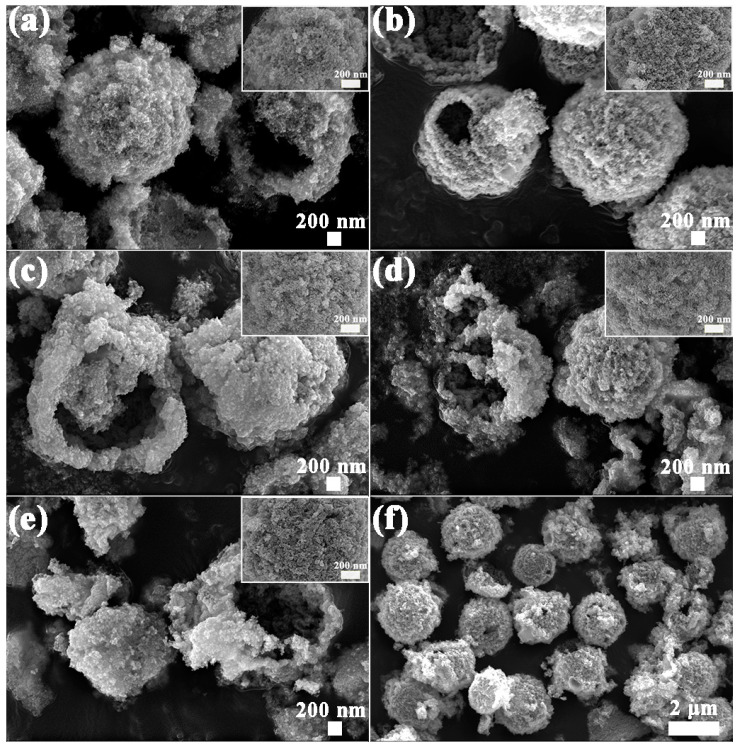
SEM images of S1–S5: (**a**) S1, (**b**) S2, (**c**) S3, (**d**) S4, (**e**) S5. (**f**) SEM image of S4 sample under low magnification.

**Figure 5 nanomaterials-12-01881-f005:**
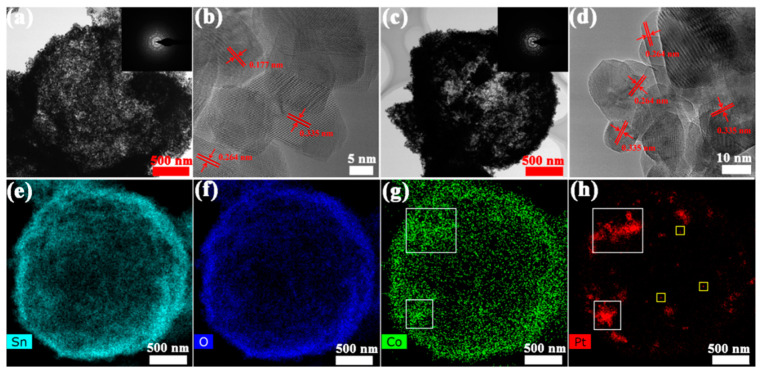
(**a**,**b**) TEM and HRTEM images of S1 (pure SnO_2_). (**c**,**d**) TEM and HRTEM images of S4. (**e**–**h**) EDS elemental mapping images of Sn (**e**), O (**f**), Co (**g**) and Pt (**h**).

**Figure 6 nanomaterials-12-01881-f006:**
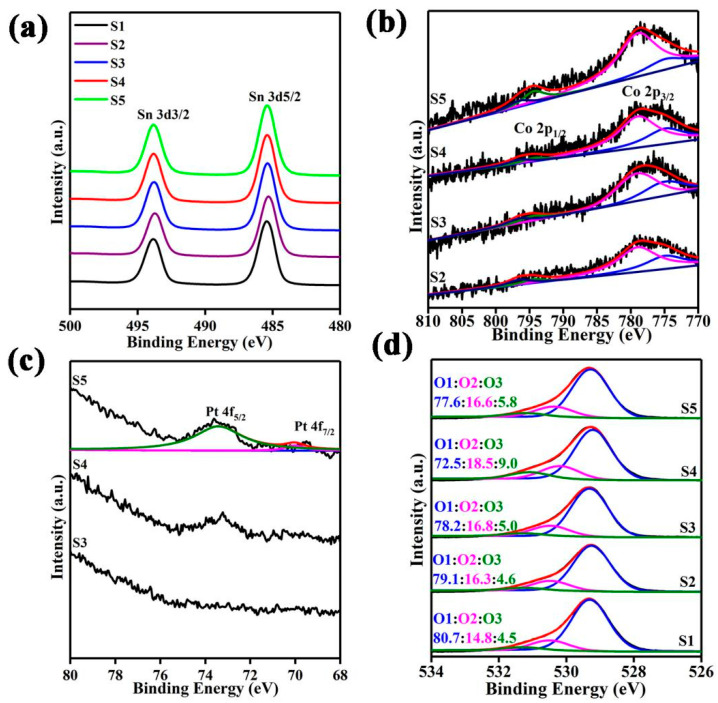
High-resolution XPS spectra of (**a**) Sn 3d, (**b**) Co 2p, (**c**) Pt 4f and (**d**) O 1s in S1–S5.

**Figure 7 nanomaterials-12-01881-f007:**
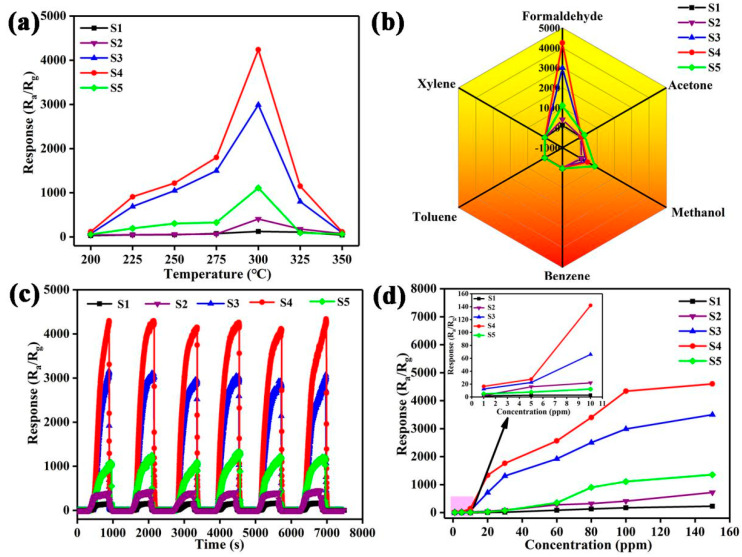
Sensing performances of S1–S5: (**a**) Response to 100 ppm formaldehyde in the temperature range of 200–350 °C. (**b**) Response to 100 ppm of interfering analytes at 300 °C. (**c**) Six-cycle dynamic formaldehyde sensing characteristics at 300 °C. (**d**) Response toward different concentration of formaldehyde.

**Figure 8 nanomaterials-12-01881-f008:**
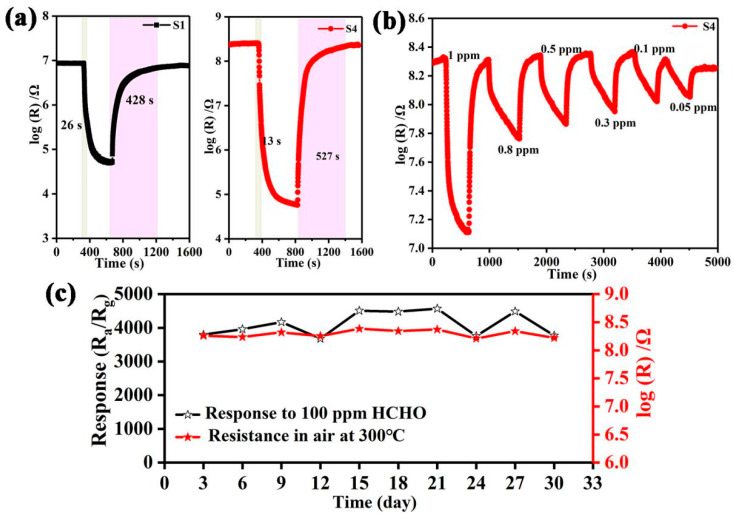
(**a**) Dynamic resistance transition of S1 and S4 toward 100 ppm of formaldehyde at 300 °C. (**b**) Dynamic resistance transition of S4 sensor (50 ppb–1ppm) at 300 °C. (**c**) Fluctuation of resistance and response values within 30 days of S4 toward formaldehyde.

**Figure 9 nanomaterials-12-01881-f009:**
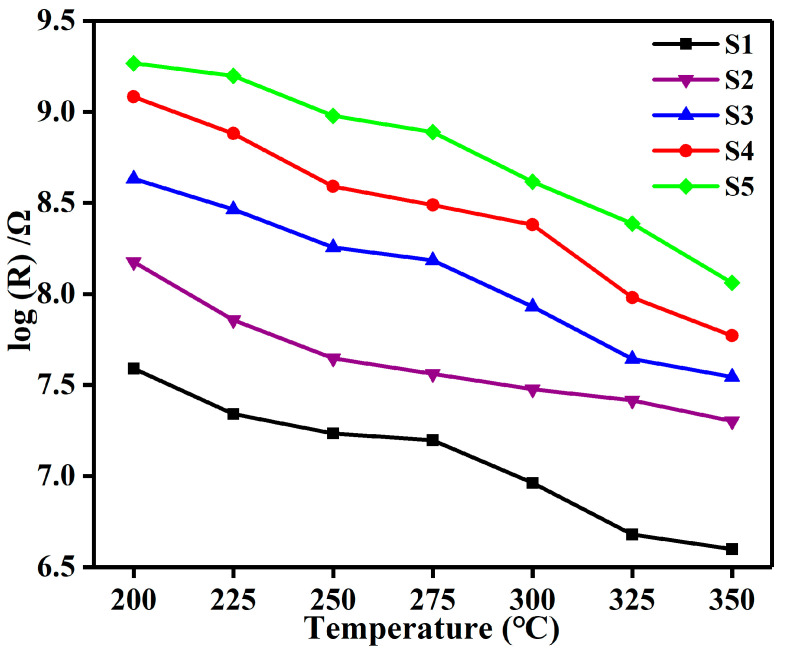
Resistance in air (log R_a_) of S1–S5 in the temperature range of 200–350 °C.

**Figure 10 nanomaterials-12-01881-f010:**
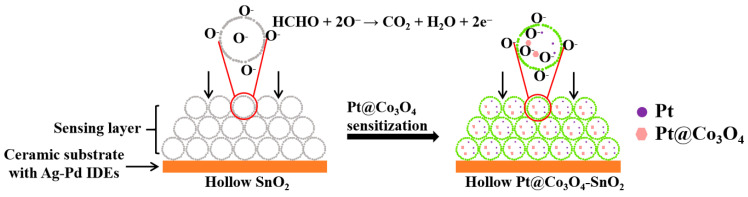
Schematic diagram of the sensing principle of SnO_2_ and Pt@Co_3_O_4_-SnO_2_ hollow microspheres.

**Table 1 nanomaterials-12-01881-t001:** Comparison of formaldehyde sensing properties in this work with other SnO_2_-based sensors.

Materials	Concentration (ppm)	Temperature (°C)	Response (*R_a_*/*R_g_*)	Limit of Detection (ppm)
Pd/SnO_2_ [9]	100	160	18.8	5
Pt/SnO_2_ [12]	1	200	16.08	0.06
Pt/SnO_2_ [13]	1	350	4.5	--
SnO_2_ [41]	100	220	30	1
SnO_2_ [42]	50	180	79	1
SnO_2_ [43]	100	120	57	0.5
NiO/SnO_2_ [44]	100	200	27.6	0.13
Zn-SnO_2_ [45]	100	400	25.7	0.5
In_2_O_3_/SnO_2_ [46]	10	275	8.7	0.5
SnO_2_/rGO/Au [47]	100	200	32.7	1
Pt@Co_3_O_4_-SnO_2_ (this work)	100 1	300	4240 16.5	0.05

## Data Availability

Not applicable.

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
