# Peer review of "Rational Design of SnO2 Hollow Microspheres Functionalized with Derivatives of Pt Loaded MOFs for Superior Formaldehyde Detection"

_nanomaterials, 2022, doi:10.3390/nano12111881_

Round 1

Reviewer 1 Report

The article describes the use of a Pt/ZIF nanomaterial in a solvothermal synthesis to form a modified SnO2 gas sensing material. A major issue with the paper (as received) is that many of the figures are near-illegible. I will raise where these cause specific issues below. I also have some concerns about the data interpretation, and again raise these below. If these are satisfactorily amended the article may be publishable in Nanomaterials.

1) Insufficient experimental detail is included. For XPS analysis, were the samples charge compensated? If so, how? Were the samples etched prior to analysis? What software was used for data analysis?

2) When discussing the preparation of the sensor devices, I could well-imagine that a simple brushing of the precursor matrix could lead to a large variance in the properties of the resultant sensor (baseline resistance, response). A demonstration of the reproducibility of sensor fabrication, leading to an estimation of the reproducibility/precision of sensor responses should be presented.

3) XRD are not spectra; diffraction patterns or diffractograms are appropriate descriptions.

4) I cannot see from Figure 3 the difference in porosity in Samples 2-5.

5) Due to the low figure quality it is hard to assess the XPS data, however it appears to me that the FWHM values for peaks associated with Co2+ and Co3+ are significantly different. What would be the justification for this?

6) I am assuming that 'oxygen deficient' in the context of XPS means oxygen vacancy. I am not sure how you can detect something that it not there?

7) To be able to reliably fit the O1s data very careful fitting is required. Again due to the low figure quality it is hard to tell, but it appears that the O1s FWHM for different peaks are very different - the smaller peaks appear to have MUCH smaller FWHM than the lattice oxygen peak. I can't immediately think of a justification for this. It seems more likely that there is only a lattice peak and one other peak, but it is hard to assess as a referee at the moment. This point is very important as it is referred to several times in the discussion as justification of the data. At the moment I remain to be convinced.

Author Response

Thank you very much for your careful review and constructive suggestions with regard to our manuscript “Rational design of SnO2 hollow microspheres functionalized with derivatives of Pt loaded MOFs for the superior formaldehyde detection ” (nanomaterials-1718551). Those comments are helpful for us to revise and improve our paper. We have studied the comments carefully and tried our best to revise and improve the manuscript and made many changes in the manuscript according to the reviewer’ good comments. Revised portion is marked in red in the paper. The main corrections in the paper and the response to the reviewer’s comments are as following. We appreciate for reviewers’ warm work earnestly, and hope that the corrections will meet with approval. Please feel free to contact us with any questions and we are looking forward to your consideration.

Yours sincerely,

Guodong Wang and Lanlan Guo

Reviewer 2 Report

The authors demonstrated that noble metal nanoparticles encapsulated in porous materials showed high sensitivity to formaldehyde even below ppm level. The authors applied MOF-templated nanoparticle synthesis to fabricated well defined and separated small enough nanoparticles in the porous materials, resulting in high sensitivity. This manuscript gives a guideline to fabrication of highly sensitive sensing materials. However, overall manuscript is now well written. The reviewer do not recommend its publication to Nanomaterials unless manuscript will be revised well.

The reviewer points out several examples below, but there are many lines that need to be revised throughout manuscript, including not limited to the following examples.

1) Co3O4 shows an important role for the selectivity to formaldehyde as discussed in the paragraph starting from line 303; however, the authors did not mention in the Introduction. Additionally, the authors did not demonstrate the sensing performance of the Pt@SnO2 without Co3O4. The authors should address this point.

2) Materials and Methods section: The authors should totally revise this section and carefully check English. For example,

  • Do not start the sentence from the Arabic numbers.
  • After a while => the authors should add time.
  • 2.2 & 2.3, “centrifugation and washing”: speed of centrifugation and the solvent for washing are missing.
  • 2.3, second paragraph is unclear. The authors described “microspheres were prepared through the SAME experimental procedure.” Which procedure?

3) Material characterizations are performed using various methods; however, the authors did not describe well and did not discuss well. For example,

  • Line 222, XPS: the authors described “The discussion and analysis of these three oxygen species will be given in the mechanism section.” There is no mechanism section. Additionally, in the paragraph starting from line 337, the authors only described high percentage of oxygen deficient (O2) in the series of S1-S4, but the authors did not show how high. From Figure 6d, it is not clear enough to identify the percentage. It can NOT be concluded from Figure 6d.
  • Line 154: “compared with the solid structure of ZIF-76” The authors should provide the SEM of ZIF-67.

The journal Nanomaterials is not sensor journal, but material journal, therefore, the authors should clearly describe the materials characterization.

4) Sensing performance: The authors showed the signal response to 50 ppb formaldehyde and claimed the LOD decreases to 50 ppb. In Figure 7d, the region shows almost 0 response. Additionally, in Figure 7b, other chemicals show higher responses compared to the response to 50 ppb formaldehyde shown in Figure 8b. The authors should clearly mention this point.

5) Introduction section: The reviewer cannot understand why the authors used Pt but not Au, Pd or other noble metal.

Author Response

(The authors gave the same response as above.)

Round 2

Reviewer 1 Report

The authors have made significant edits to the original submission, the figure quality is much improved the data analysis now seems sound. I am happy for the article to be published 'as is'

Author Response

Thank you very much recognized by the peer reviewers

Reviewer 2 Report

The authors have met the reviewer's comments and well described the detail material characterizations as well as sensing performance. The reviewer found that the revised manuscript is well written including Introduction except English language and style. The reviewer would like to recommend to publishing in Nanomaterials with some minor revisions including English as follows.

English language and styles: For example, in Methods section, the reviewer did not say that the authors do not start the "paragraph" with Arabic numbers but do not start the "Sentence" with Arabic numbers.
e.g., Line 93, "582 mg Co(NO3)2•6H2O and ..." => "Co(NO3)2•6H2O (582 mg) and ..."
The reviewer found some non-necessary spaces (e.g., line 185), non-necessary period (e.g., line 189), some missing spaces before unit (e.g., line 148) and so on.

The reviewer still does not understand the importance of cobalt oxide. The reviewer understood the semiconducting properties of cobalt oxide and the importance of improving sensing performance. However, comparing PtSnO2 with and without Co3O4 in Table 1, the limit of detection levels are not big difference (60 ppb in Ref. 12 vs 50 ppb in this work). The authors should describe more on this point.

Author Response

English language and styles: For example, in Methods section, the reviewer did not say that the authors do not start the "paragraph" with Arabic numbers but do not start the "Sentence" with Arabic numbers.

e.g., Line 93, "582 mg Co(NO3)2•6H2O and ..." => "Co(NO3)2•6H2O (582 mg) and ..."

The reviewer found some non-necessary spaces (e.g., line 185), non-necessary period (e.g., line 189), some missing spaces before unit (e.g., line 148) and so on.

Responds: Thank you very much for the professional comments on our manuscript. We appreciate your careful reading of our manuscript. Firstly, we have revised the  the sentence which started with Arabic numbers in the Materials and Methods section at Page 2, Line 95-97, Page 3, Line 103-105 and 111-113. Secondly, we have checked the non-necessary, non-necessary period, and missing spaces in the revised manuscript at Page 3, Line 108, 150, and Page 5, Line 192-193 and so on.

The reviewer still does not understand the importance of cobalt oxide. The reviewer understood the semiconducting properties of cobalt oxide and the importance of improving sensing performance. However, comparing PtSnO2 with and without Co3O4 in Table 1, the limit of detection levels are not big difference (60 ppb in Ref. 12 vs 50 ppb in this work). The authors should describe more on this point.

Responds: Thank you very much for the professional comments on our manuscript. We appreciate your careful reading of our manuscript. The importance of cobalt oxide can be explained as follows. On one hand, from Figure 7a-d, the incorporation of cobalt oxide can indeed improve the sensing performance of SnO2 hollow spheres because of the p-n heterojunctions between p-Co3O4/n-SnO2. On the other hand, the cobalt oxide is derived from ZIF-67, which is used as templates to obtain ultra-small Pt nanoparticles (Pt@ZIF-67). As can be seen from Figure 7, ultra-small Pt nanoparticles obtained together with Co3O4 also play a important role in enhancing the sensing performance of SnO2.

It seems that the limits of detection of Ref. 12 (60 ppb) and this work (50 ppb) are not big difference. However, compared with the doping amount (1 wt%) and particle size of Pt (>5 nm) in the Ref. 12, the sensing properties of Pt@Co3O4-SnO2 toward formaldehyde are significantly enhanced even by the quite low Pt loading (~ 0.3 wt%) due to the finer dispersion of Pt (~3 nm) attained by using Pt@ZIF-67. Therefore, the sensor fabricated in this work is more meaningful for practical application eventually due to the expensive price of noble metals. The related statement is in the revised manuscript at Page 9, Line 316-322 and Page 10, Line 323-324.
